# The Heterogeneous Multiple Sclerosis Lesion: How Can We Assess and Modify a Degenerating Lesion?

**DOI:** 10.3390/ijms241311112

**Published:** 2023-07-05

**Authors:** Olivia Ellen, Sining Ye, Danica Nheu, Mary Dass, Maurice Pagnin, Ezgi Ozturk, Paschalis Theotokis, Nikolaos Grigoriadis, Steven Petratos

**Affiliations:** 1Department of Neuroscience, Central Clinical School, Monash University, Melborune, VIC 3004, Australia; olivia.ellen1@monash.edu (O.E.); sining.ye1@monash.edu (S.Y.); danica.nheu1@monash.edu (D.N.); dassmaryparimala@gmail.com (M.D.); maurice.pagnin@monash.edu (M.P.); ezgi.ozturk@monash.edu (E.O.); 2Laboratory of Experimental Neurology and Neuroimmunology, Department of Neurology, AHEPA University Hospital, Stilponos Kiriakides Str. 1, 54636 Thessaloniki, Greece; ptheotokis@auth.gr (P.T.); grigoria@med.auth.gr (N.G.)

**Keywords:** multiple sclerosis, oligodendrocyte, myelination, inflammation, microglia, macrophage, biomarkers, imaging diagnostics

## Abstract

Multiple sclerosis (MS) is a heterogeneous disease of the central nervous system that is governed by neural tissue loss and dystrophy during its progressive phase, with complex reactive pathological cellular changes. The immune-mediated mechanisms that promulgate the demyelinating lesions during relapses of acute episodes are not characteristic of chronic lesions during progressive MS. This has limited our capacity to target the disease effectively as it evolves within the central nervous system white and gray matter, thereby leaving neurologists without effective options to manage individuals as they transition to a secondary progressive phase. The current review highlights the molecular and cellular sequelae that have been identified as cooperating with and/or contributing to neurodegeneration that characterizes individuals with progressive forms of MS. We emphasize the need for appropriate monitoring via known and novel molecular and imaging biomarkers that can accurately detect and predict progression for the purposes of newly designed clinical trials that can demonstrate the efficacy of neuroprotection and potentially neurorepair. To achieve neurorepair, we focus on the modifications required in the reactive cellular and extracellular milieu in order to enable endogenous cell growth as well as transplanted cells that can integrate and/or renew the degenerative MS plaque.

## 1. Introduction

Multiple sclerosis (MS) pathology presents as several multifocal lesions characterized by inflammation and demyelination within the central nervous system (CNS) [1]. These pathological hallmarks reported during the initial stages of the disease are often accompanied by the clinical manifestation of functional relapses followed by remission, also known as relapsing–remitting multiple sclerosis (RRMS). Hence, the heterogeneity of this disease governs the course and severity of the neurological outcomes in patients with MS [2,3]. Although the exact cause is unknown, the infiltration of peripheral inflammatory lymphocytes and myeloid cells degenerate the myelin sheath along axons [4,5,6,7]. The destruction of myelin deposits biological debris in the extracellular milieu within the CNS, further resulting in axonal dystrophy, synaptic injury, and gliosis, as observed in patients who transition into the progressive stage of MS [1,8,9,10].

The severity and frequency of relapses during the RRMS phase can be alleviated to some extent by currently available disease-modifying treatments (DMTs) such as fingolimod, which modulates immune cell trafficking. Despite the significant therapeutic impact of DMTs in improving the symptoms of RRMS, the conversion to secondary progressive MS (SPMS) remains a prognostic outcome with further neurological progression governed by the lack of treatment options. Therefore, treatment for progressive MS remains a major unmet medical need [11,12,13,14].

The urgent need for novel and/or repurposed therapeutics has been highlighted by a recent retrospective study conducted in Iran that predicted 93% of RRMS patients would transition into SPMS within a span of 30 years despite receiving early intensive DMTs or escalation treatment [15]. Another clinical study conducted by Kalincik et al. showed a smaller proportion of RRMS patients advancing to the progressive stage with the administration of DMTs compared to the control group, with no significant effect from DMTs in progressive MS patients [16]. Therefore, to address the gap in the treatment of progressive MS, potential therapies that attempt to limit demyelination of axons and restore axonal function may have a positive therapeutic outcome in ameliorating MS chronic disease progression.

In this review, we discuss the comprehensive cellular and molecular mechanisms involved in the progressive stages of MS, with particular focus on the role of phagocytosis in promoting remyelination and axonal regeneration within the CNS. Moreover, we highlight the benefits of using cellular delivery strategies, such as genetically modified hematopoietic stem cells (HSCs) that can express potential therapeutic proteins capable of crossing the blood–brain barrier (BBB) to repair the CNS environment during progressive MS. Furthermore, we address the lack of cellular and molecular biomarkers used in MS diagnosis and in the assessment of progression by exploring novel potential biomarkers as a means of monitoring neuroreparative strategies designed for clinical translation during trials of recruited individuals living with SPMS.

## 2. Pathogenesis

### 2.1. Alterations in the Blood–Brain Barrier during MS

Structurally, the BBB is a tightly regulated biophysical cellular barrier consisting of cerebral endothelial cells, pericytes, and a basement membrane that interacts with astrocyte end-feet (for review, see [17]). This barrier separates the CNS from the systemic circulation and can selectively restrict molecules and cells from passively entering the CNS environment (for review, see [17]). Due to these non-permissive barriers imposed by the tight junctions of the brain vascular endothelial cells, a key feature required when designing drug delivery to the CNS is to overcome the sealed lateral membrane inter-endothelial cleft of the BBB. In RRMS, abnormalities in tight junctions located in the blood vessels of active lesions can cause a temporary disruption to the BBB [18,19]. This alteration to the BBB is accompanied by modulation of the vascular endothelial cytoskeleton and the tight junctions, attributed to the release and accumulation of pro-inflammatory cytokines secreted from differentiated T cells and activated macrophages [20]. Additionally, the interaction between endothelial cells and monocytes produces reactive oxygen species (ROS), which further contribute to decreased BBB integrity, consequently making it more susceptible to increased migration and infiltration of inflammatory cell-trafficking into the CNS [21,22] and potentially promulgating axonal and neuronal damage [23].

Other pathogenic factors that are upregulated within the extracellular milieu include matrix metalloproteinases (MMPs), which regulate the cleavage of the extracellular matrix (ECM) proteins and penetrate and mobilize innate and adaptive leukocytes throughout the CNS tissue [24]. The importance of these findings of the pathogenic involvement of MMPs during inflammatory demyelination relates to the propagation of perivascular cells and microglia expanding neuroinflammatory lesions [24,25].

Throughout the neuroinflammatory lesions, the cytokines generated by differentiated T cells that can drive MMP activity within the pathogenic milieu can include tumor necrosis factor-alpha (TNF-α) and interleukin (IL)-17, which can stimulate MMP-9 transcription to degrade the ECM [26]. This would imply that a mutual promotion of MMP activity and recruitment of immune cells exists [26]. Intriguingly, Kanesaka et al. previously reported that MMP-3 serum levels were higher during the relapsing stage of MS than in remission [27], implicating MMP-3 as a molecular biomarker for the acute phase of inflammatory demyelination.

The relationship of MMP activity during lesion formation in active MS demonstrates a dysregulation of MMP-1, -2, -3, -7, and -9 expression by demyelinating macrophages. When these lesions progress to the chronic state, macrophages display a downregulation of MMP expression [28,29]. Alexander et al. found that the decrease in MMP-8 and MMP-9 serum levels correlated with a reduction in the number of gadolinium-enhancing magnetic resonance imaging (MRI) lesions [30], suggesting that the identification of reduced levels of these MMPs may be an appropriate marker that defines neurodegenerative change during progression. Consequently, cellular-mediated alterations of the BBB may provide clues to identifying the principal cells driving MS pathology and, therefore, may assist in developing cell-based therapeutic vehicles to limit BBB permeability and immune cell infiltration.

### 2.2. Experimental Evidence for Autoreactive Proinflammatory T-Cell Entry across the BBB during MS

Immune cell infiltration across the BBB may well be the primary target for future therapeutics to limit pathogenic T-cell subsets from perpetuating the cycle of neuroinflammation. Recent evidence has implicated the dual immunoglobulin domain-containing cell adhesion molecule (DICAM) as a prominent receptor expressed on invading T-helper (Th) cell 17+ lymphocytes during RRMS and progressive MS [31]. Importantly, these investigators identified the ligand for DICAM, the αvβ3 integrin, to be expressed in the BBB endothelial cells during inflammatory disease activity in MS demyelinating lesions identified from archival tissue. DICAM is a member of the CTX family of adhesion proteins, of which the extracellular Ig domain 2 was unrelated to the arginyl–glycyl–aspartate (RGD)-binding domain of the αvβ3 integrin [32]. The binding of DICAM expressed on CD4+ T cells to the αvβ3 integrin expressed on endothelial cells at the BBB can promote the inhibition of Akt signaling, leading to integrin β3/focal adhesion kinase (FAK) activity with resultant actin cytoskeletal rearrangement [33] to transiently open up the tight junctions, allowing for transendothelial cell migration of the Th17+ pathogenic lymphocytes [31]. Indeed, by antagonizing DICAM interaction with endothelial αvβ3 integrin, it was demonstrated that the clinical course of experimental autoimmune encephalomyelitis (EAE) could be reduced in severity, which correlated with a reduction in the transendothelial cell migration across the BBB [31]. The data suggest that DICAM plays an integral role in the trafficking of pathogenic Th17 cells across the BBB during active inflammatory demyelinating lesions in MS.

### 2.3. Outside-in vs. Inside-out Hypotheses Characterizing the Pathogenesis of MS 

Currently, there are two competing hypotheses that explain the etiopathogenesis of MS: the *inside-out* and *outside-in* models, which manifest similar clinical presentations (for review, see [34]). These models operate simultaneously in MS patients to form inflammatory demyelinating lesions (Figure 1). The *outside-in* model can be described as dysregulation of peripheral immune cells that results in an autoimmune attack against myelin within the CNS [35,36,37]. Classically, myelin antigen-activated immune cells such as circulating T lymphocytes, B lymphocytes, and activated monocytic cells traverse the disrupted BBB and infiltrate the CNS to target putative myelin epitopes (for review, see [38]). Although this hypothesis highlights the prominent role of immune cells promulgating inflammation and demyelination in MS, it fails to explain the underlying cause that initiates the autoimmune response [2]. Investigational inference may be derived from a recent genome-wide analysis mapping the X chromosome from 47,429 MS patients, which demonstrated that gene variants for MS susceptibility were enriched in human microglia, along with peripheral immune cells that include the C-type lectin-like protein (CLECL1), as an example [39].The CLECL1 protein is important for self-recognition, with its 20-fold reduction in MS cortical tissue highlighting that microglia may be central to autoimmune-dependent demyelination. This study suggests that both the adaptive immune response and resident glial cells play a collaborative role in MS etiopathogenesis. Therefore, the *outside-in* model, which primarily supports the role of the adaptive immune system in propagating ongoing inflammation, cannot be the sole explanation for the heterogeneous pathogenesis that manifests in MS. In contrast, the *inside-out* hypothesis maintains that the pathology of MS is initiated and, importantly, progresses, as a result of the degeneration of oligodendrocytes in the CNS governed by altered signaling of microglia and astrocytes, which is responsible for innate immunity [40,41]. The slow degradation of oligodendrocytes may cause antigenic myelin proteins that are citrullinated to be released into the circulation over time, which in turn activates T- and B-lymphocytic responses that perpetuate an inflammatory cycle [42,43,44].

Recent data by Bjornevik et al. provided evidence that EBV infection may be a leading cause of MS, but the exact mechanism for disease initiation is still unclear [45]. Possible mechanisms may include the activation of autoreactive B cells and molecular mimicry, in which CD4^+^ T cells are able to recognize both EBV and myelin peptides [46,47,48]. However, these mechanisms are still not well-defined during MS pathogenesis. Serafini et al. highlighted that EBV infection was evident in the B cells and plasma cells infiltrating into the brain of MS patients, but the cause of the migration of infected B cells to the site of injury in the CNS was not explored [49]. Furthermore, Bjornevik et al. showed that the median time from EBV seroconversion to MS clinical manifestations was 7.5 years [45]. The studies suggest that EBV may not be the primary cause of MS and that there is another cofactor that triggers the initiation of the disease.

Ermin is a component of myelin that contributes to the integrity of myelin sheaths, and its expression has been shown to be downregulated in mouse brains during cuprizone-mediated demyelination as well as mutations identified from isolated peripheral immune cells of individuals from one family that exhibited RRMS [40,50,51]. Bis-cyclohexanone-oxaldihydrazone (Cuprizone) is a copper chelator that, when given in the feed of rodents for as little as 3 weeks, promotes the selective dystrophy of mature oligodendrocytes and robust demyelination in the corpus callosum along with the dorsal fornix [52]. Cuprizone may downregulate the expression of Ermin, leading to a loss of myelin integrity and disruption of the oligodendrocyte cytoskeletal architecture [53]. Ziaei et al. described the loss of myelin sheath integrity in 5-month-old Ermin knockout (KO) mice and axonal damage in 3-month-old Ermin KO mice [40]. Microgliosis, astrogliosis, oligodendrocytes loss, and upregulation of inflammatory response genes were also observed in 8-month-old Ermin KO mice [40]. In addition, induction of EAE in Ermin KO mice resulted in a higher number of CD45^HI^ infiltrating monocytes in the CNS compared to EAE-induced wildtype (WT) mice [40], providing compelling evidence for the inside-out paradigm of MS etiopathogenesis. However, the relationship between Ermin and MS etiopathogenesis has not been thoroughly explored enough to establish causal relationships with oligodendrocyte–myelin dysfunction.

### 2.4. The Role of Macrophages and Microglia in Driving Neurodegeneration in MS

Macrophages and microglia, arising from hematogenous and endogenous precursors, are important respondents in MS immunopathogenesis [54]. Microglial cell precursors are established prior to birth and can undergo self-proliferation in situ [55]. A study by Ajami et al. showed that blood monocytes gave rise to infiltrating macrophages within the CNS but not microglia, and these infiltrating monocytes were associated with the progression of EAE [56], implying that endogenous microglia and monocyte-derived macrophages are two separate populations. Furthermore, the infiltrating monocytic-derived macrophages at the peak of EAE were present transiently in the CNS, whereas proliferating microglia were still detected 3 months later [56]. Mildner et al. described peripheral Ly-6C^hi^CCR2^+^ monocytes infiltrated into the CNS and differentiated into resident microglia but only when the brain and spinal cord of cuprizone or healthy mice were irradiated, highlighting the need for brain conditioning and BBB disruption before engraftment [57]. 

Human monocytes can be divided into CD14^+^CD16^−^ classical, CD14^+^CD16^dim^ intermediate, and CD14^dim^CD16^+^ non-classical monocytes [58]. In mice, the CCR2^+^CD62L^+^ CX3C motif chemokine receptor 1 (CX3CR1)-EGFP^lo^Ly6C^+^ monocytes correspond to human classical monocytes and CCR2^−^CD62L^−^CX3CR1-EGFP^hi^Ly6C^lo^ monocytes correspond to human non-classical monocytes [59]. A fate-mapping analysis of the monocyte origin revealed that classical Ly6C^+^CX3CR1^int^ monocytes from bone marrow and blood give rise to nonclassical Ly6C^−^CX3CR1^hi^ monocytes in both compartments [60]. It is argued that a more robust measure of assessing classical and non-classical monocytes is to incorporate further markers such as CD43 and Ly6c to discriminate between these two phenotypes [59]. CD43 and Ly6c were substantiated to be better monocyte markers in CX3CR1^+/EGFP^ mice because the expression of the green fluorescent protein (GFP), a proposed marker of CX3CR1 levels in these transgenic mice, in fact did not accurately reflect the actual monocytic/microglial cell response [61]. In the CNS, chemokine (C-X3-C motif) ligand 1 (CX3CL1) is mostly expressed in neurons, whereas CX3CR1 expression is detected in the microglia [62,63]. The deficiency of CX3CR1 in cuprizone-fed mice was shown to lower triggering receptor expressed on myeloid cells 2 (TREM2) and CD11c expression in microglia, leading to lower phagocytic activity and clearance of myelin debris [64].

Similar to T lymphocytes, different cytokines and chemokines can induce macrophages and microglia to release either pro-inflammatory cytokines TNF-α and IL-1β to promote a pro-inflammatory environment or anti-inflammatory cytokines IL-10 and transforming growth factor (TGF)-β to stimulate regeneration and repair [65,66]. Interestingly, in active demyelinating white matter lesions in MS-diagnosed individuals, macrophages are observed to co-express the proinflammatory marker CD40 (commonly expressed by antigen-presenting cells) and anti-inflammatory marker CD206 (commonly expressed by phagocytic macrophages), indicating an intermediate activation status of macrophages [67]. However, in chronic active lesions, macrophages and microglia in slow-expanding edges co-expressed CD40 and the pro-inflammatory scavenger marker CD163 [67]. The mixed profile of macrophages governing the inflammatory demyelinating lesion in MS suggests that these cells are in constant transition between proinflammatory and anti-inflammatory states during MS pathophysiology and may be very challenging to target to resolve chronic active lesions. Moreover, this depicts the heterogeneity of macrophages and microglial cell polarization and highlights the complex classification of microglia and macrophage phenotypes in MS lesions at different stages of the pathogenic pathway that can be classified with deeper transcriptional profiling to identify their physiological relevance.

During inflammatory disease, macrophages and microglia attempt to maintain compartmentalization of homeostasis through plasma membrane polarization to expedite phagocytosis of accumulated myelin debris and apoptotic cells within the extracellular milieu, a pathogenic mechanism that results in physiological inhibition to promote neurorepair during MS [64,68,69]. Under the controlled neuroinflammatory environment of EAE, Epstein et al. demonstrated that myelin uptake is dependent on receptor-mediated endocytosis, with evidence of myelin lamellae binding to the coated pits of macrophages, where a high concentration of ligand–receptor-binding interaction is prevalent prior to endocytosis [70]. During the pathogenesis of MS, several receptors are expressed on macrophages/microglia that may be involved in the uptake of myelin, namely, Fc receptors, complement receptors (CR), and scavenger receptors (SR) [71,72,73,74]. A pioneering study by Ulvestad et al. demonstrated that phagocytic cells in the parenchyma and perivascular region of active MS lesions strongly expressed Fc receptors (FcR) FcRI, FcRII, and FcRIII, whereas microglia expressed fewer FcR in normal-appearing white matter (NAWM) [75]. The uptake of myelin is also based on opsonization and epitope recognition by antibodies [76]. There exist data to suggest that circulating anti-myelin basic protein (MBP) antibodies isolated from the sera of individuals living with MS can act as hydrolyzing enzymes against histone proteins with high rates of H1 cleavage activity to damage nuclei of cells [77]. These auto-reactive IgGs, defined as abzymes, have been demonstrated to hydrolyze up to 17 H1 histones, and the catalytic antibodies have been associated with expanding disability during MS progression [78], suggesting at the very least that they may be a good biomarker for MS progression. However, it is not clear how these antibodies may promote cell death during MS, but anti-DNA hydrolyzing abzymes isolated from another autoimmune condition, systemic lupus erythematosus (SLE), show cytotoxicity when incubated with the cultured L929 immortalized adipose cell line [79]. However, the relevance of myelin-specific antibodies in the pathogenesis of MS remains to be elucidated, as they are not only found in MS patients but are also present in the circulation of healthy subjects [80,81].

Another receptor responsible for myelin internalization is CR, as confirmed by Loveless et al., who discovered several phagocytic cells expressing CR in progressive MS lesions [82]. An active subtype, CR3, was observed to facilitate approximately 80% of myelin clearance; however, without the presence of an active complement, the myelin clearance rate dropped to 55–60% [74]. These results highlight how overactivated CR3 can contribute to myelin clearance. Additionally, in a study that induced EAE in SR-A^−/−^ mice, demyelination in the CNS and disease severity were significantly reduced compared to the wild-type control [83]. In the rim of chronic active MS lesions, SR-A1/II was also found in foamy phagocytes and ramified microglia, suggesting receptor involvement in the early phase of myelin uptake and demyelination [73]. Therefore, understanding the function of specific microglial receptors during various stages of the disease may enable the modification of these receptors to be used as a therapeutic strategy for increasing myelin debris clearance and promoting anti-inflammatory environment in the lesion sites.

Another receptor, MER tyrosine–protein kinase (MERTK), is a part of the TAM family of receptors and can regulate myelin phagocytosis in human monocyte-derived macrophages and microglia [84]. The TAM receptors consist of TYRO3, AXL, and MERTK with their known ligands, growth arrest-specific 6 (GAS6) and protein S [85,86]. The TAM family of receptor tyrosine kinases has been shown to play a role in cell proliferation and survival, regulation of the immune system, and phagocytosis of cells [87]. In a study by Binder et al., cuprizone-induced demyelination in GAS6^−/−^ mice was more severe compared to wild-type mice [88]. Additionally, Weinger et al. identified that levels of soluble MERTK were higher in chronic active MS lesions, whereas in chronic silent MS lesions, the level of soluble AXL was elevated compared to healthy controls [89]. Furthermore, elevated levels of soluble AXL and MERTK receptors were accompanied by low levels of GAS6 in MS lesions, resulting in the dysregulation of GAS6/TAM signaling and possibly prolonged lesion activity [89]. These studies also imply that polymorphism in the genes of TAM receptors or their ligands may be associated with an increased risk of MS. In a candidate gene study conducted by Ma et al., 12 single nucleotide polymorphisms (SNPs) were found in the MERTK gene, which is associated with MS susceptibility [87]. In a supporting study, Shen et al. established that MERTK was required for activation of microglia, microglial phagocytosis, and remyelination [90]. A recent study showed that TREM2 was expressed in MS lesions and that TREM2 on microglia facilitated myelin debris clearance in cuprizone-fed mice [91]. TREM2 activates downstream spleen tyrosine kinase (SYK) via phosphorylation of DNAX-activating protein of 12 kDa (DAP12) to promote microglia activation and phagocytosis of cellular debris (for reviews, see [92,93]). Therefore, phagocytic activity may indeed regulate the effectiveness of myelin clearance to enable newly formed myelin to repair denuded axons. 

However, a fundamental problem in neuropathology exists when attempting to resolve the contribution of peripheral macrophages versus endogenously activated microglia to the expanding MS lesions that exist in the CNS of people that have lived with MS for protracted periods of time with variable relapse rates and stages of progression. Phenotyping peripherally derived macrophages and endogenously activated microglia poses challenges to investigators defining pathogenesis and outlining therapeutic strategies that can limit neurodegenerative changes orchestrated by the activities of these cells. An important transmembrane protein TMEM119 and the purinergic receptor P2RY12 have been successfully used as markers to differentiate resident microglia from peripheral macrophages [94,95,96]. In the gray matter, the numbers of P2RY12^+^ TMEM119^+^ microglia in demyelinating lesions were similar to in normal-appearing gray matter (NAGM), possibly due to less stimulation from locally generated cytokines, a consequence of fewer infiltrating lymphocytes as compared to inflammatory demyelinating white matter [97]. However, it was proposed that the morphological differences exhibited by microglial subtypes can be used to differentiate their unique characteristics at specific brain regions. In the white and gray matter of archival control brain tissue, microglia display a small soma with slender ramified and elongated fine processes. Whereas in active white matter lesions, the microglia displayed enlarged amoeboid and thick ramified phenotypes, in the gray matter, some microglia were rod-shaped [97]. The function of rod-shaped microglia remains elusive, but synaptic stripping has been proposed [98,99], which may be one mechanism that drives the rate of demyelination throughout the gray matter. Since the white matter and gray matter microglia may express distinguishable phenotypes throughout brain lesions of individuals with MS, identifying specific microglial cell populations and their respective pathogenic roles during the disease course are important to determining potential targets for disease chronicity and progression.

In support of this rationale, transcriptomic analysis of microglia has revealed genes that were not only expressed at region-specific sites but also that were disease-specific [100]. TMEM119 and P2RY12 are expressed in the population of homeostatic microglia, but expression of the genes variably decreases at brain lesion sites of archival tissue from individuals that lived with MS or Alzheimer’s disease [101,102]. Moreover, Zrzavy et al. found that the increased numbers of active microglia in the NAWM of archival MS brain tissue exhibited reduced expression profiles of P2RY12 and, importantly, the gene expression was depleted in active and slow-expanding MS lesion sites. On the other hand, the expression of TMEM119 decreased in NAWM and, more notably, as the lesion progressed, indicating that there is a reduction of homeostatic signatures in microglia or, alternately, that lower numbers of microglia are present within active brain lesions [102]. The microglia isolated from the gray matter demonstrated elevated expression profiles of genes associated with the type I interferon response, whereas white matter microglia had a higher expression of genes related to the nuclear factor kappa B (NF-κB) pathway [100]. These data suggest that there exist two separate immune regulatory mechanisms that are operative among these two microglial cell subtypes. Furthermore, microglia that reside in the NAWM of individuals living with MS were enriched with genes related to lipid metabolism, the differentiation of foam cells, the lysosome, and other signaling pathways (ABHD2, LPL, ASAH1, CTSD, SCARB2). On the other hand, microglia from NAGM were enriched with genes associated with glycolysis and metal ion homeostasis (ABCB6, SDC1, CCR2, LPAR6, SLC25A37) [100]. Interestingly, the homeostatic genes of microglia in MS NAWM and NAGM were not affected [100]. The findings imply that lipid processing and iron metabolism pathways are disrupted early in microglia throughout NAWM and NAGM during MS pathogenesis. These data provide insights into which therapeutic targets could provide efficacy to limit ongoing slow-burning demyelinating lesions during MS progression.

### 2.5. Astrocyte Activity in the Evolution of the MS Plaque

Astrocytes give rise to extensive branches that develop as fibrous foot processes, attaching to the blood vessels or shaping the uniform globule located around axons and synapses of neural cells (for review, see [103]). Given the distribution of astrocytes within the CNS, they play a prominent role in orchestrating the connection between the vascular system and neural function, are associated with enhanced neuronal survival, and promote synaptic plasticity by regulating neuronal circuit activity in response to metabolic and structural support [104,105]. Intriguingly, studies have shown that astrocytes are not only neuroprotective but can also be deleterious in certain neurological disorders [106,107]. Abnormal astrocytes have been found in abundance in post-mortem brain tissue from patients with MS, Alzheimer’s disease, and Parkinson’s disease [107,108,109]. These studies suggest that astrocytes alter their function in response to neuropathological insults, especially under inflammatory conditions, which can transform neuroprotective “resting astrocytes” into neurotoxic “reactive astrocytes” [110]. Reactive astrocytes were originally defined in neuropathology as glia that respond during injury and disease to the altered CNS microenvironment, and mostly exert damage-associated responses but can also exert neuroprotection. The heterogeneity of the reactive astrocytes can be seen in their transcriptomic profiles during injury and disease to the CNS but generally respond through proliferation, hypertrophy, and synthesis of proinflammatory and chemical mediators (for review, see [111]).

It has been identified that astrocytic responses are modulated by canonical signaling pathways such as IKK/NF-κB, a classical mechanism implicated in neuroinflammation [112,113,114,115]. The NF-κB signaling pathway in astrocytes can be stimulated by various factors, including pathogenic extracellular or cellular debris, reactive oxygen species (ROS), and pro-inflammatory cytokines such as TNF-α, IL-1β, and IL-17 secreted from peripheral immune cells and resident microglia by activating on their toll-like receptors (TLRs) [114,116]. To expand these findings, Wheeler et al. found that the astrocytes of EAE-induced mice and MS patients had a reduced nuclear factor erythroid 2-related factor 2 (NRF2) expression and a higher MAFG transcription factor level, which promoted DNA methylation and inflammation in the CNS and limited anti-inflammatory activity [117]. They also suggested that granulocyte–macrophage colony-stimulating factor (GM-CSF) secreted by pathogenic T cells might promote pro-inflammatory activity of astrocytes by amplifying the MAFG/MAT2α signaling pathway in EAE-induced mice and in individuals living with MS [117]. In addition, previous studies indicated that reactive astrocyte toxicity was mediated by secreted proteins, including SPARC, C3, and lipocalin-2 reactive markers [107,118,119]. Surprisingly, these neurotoxic factors did not cause the apoptosis of oligodendrocytes within the inflammatory environment. This finding indicated that the fundamental toxic agent resulting in myelinated cell apoptosis and axonal injury had not been found [120]. However, the enrichment of apolipoprotein (APO)-E and APO-J in reactive astrocytes suggests that an altered lipid metabolism in astrocytes may mediate cell apoptosis during inflammation [120]. 

Astrogliosis-mediated synaptic phagocytosis could be a compensatory mechanism independent of microgliosis-dominant engulfment for maintaining CNS homeostasis [121,122]. Synaptic elimination can be achieved via multiple epithelial growth factor (EGF)-like domains 10 (MEGF10)/MERTK phagocytic signaling pathways in astrocytes. MEGF10/MERTK-mediated astrocytes have been observed to trim synapses, thus promoting the precise structure of neural circuits and synaptic plasticity [121]. Moreover, in MEGF10^−^/^−^ mouse models, elevated levels of neuronal apoptosis and a reduced level of astrocyte phagocytosis were observed, which suggests that astrocyte phagocytotic function can be modulated by the complement component 1q (C1q)/MEGF10 signaling pathway [121]. 

The development of single-cell (sc)-RNA sequencing techniques has uncovered significant diversity in astrocyte populations, with spatial and temporal differences under physiological and pathological stimuli, throughout the CNS [117,123]. Despite this technical advancement, several challenges have been identified in studies addressing the heterogeneity of astrocytes in health, aging, and disease. One of the major hurdles is that only limited numbers of neuropathogenic astrocytes can be collected from CNS tissues for downstream cell-signaling analysis. This is primarily due to the lack of reliable surface markers to distinguish astrocyte subpopulations, thereby preventing any comprehensive exploration of astrocyte diversity. For example, glial fibrillary acidic protein (GFAP), which is considered a classic marker of astrocytes, has not been useful in identifying neurotoxic astrocytes isolated from brain and spinal cord in MS-like animal models such as EAE [117]. Recent studies have focused on investigating the molecular characteristics of astrocytes at the level of DNA and mRNA, which may provide an alternative avenue to differentiate cell populations of interest during the course of neuroinflammatory and neurodegenerative diseases.

Recently, Clark and colleagues identified that the upregulation of the transcription factor X-box-binding protein 1 (XBP1) can promote the frequency of astrocytic cell-derived proinflammatory mechanisms exhibited during EAE and MS [124,125]. XBP1 activation is involved in processes that lead to its mRNA splicing unconventionally when potentiated by the endonuclease inositol-requiring enzyme 1 (IRE1), expressing the potential to serve as an mRNA marker for pathogenic astrocytes in MS. However, XBP1 levels determined by 3′-prime based RNA-sequencing techniques showed inconsistencies, highlighting the need to address the technical issue of sc-RNA sequencing in regard to low sensitivities of the detection of potential genetic and transcriptomic markers. Improving the read depth and enriching targeted reads during RNA sequencing may be the approach to enhancing the sensitivity. Clark et al. established the FIND-seq protocol, which enriches the population of rare astrocyte subtypes by specific DNA-barcoded beads and captured by microfluidic encapsulation [124]. This innovative strategy significantly improves the detection of gene diversity of pathogenic astrocytes in comparison to the scDrop-seq [126] and Probe-seq techniques [127].

### 2.6. Types of MS Lesions

Neuroinflammation damages the BBB and creates macroscopic plaques, whereas neurodegeneration causes injury to neurons and their axonal extensions [1,128]. Mononuclear perivenular cuffing on the lesion and the infiltration into the adjacent white matter have also been observed [24,129]. The infiltration of immune cells and activated phagocytic cells is often associated with the formation of active lesions. The activated microglial cells are commonly found at the edge of demyelinating lesions, periplaque white matter, and NAWM, whereas macrophages are found in the center of active lesions [2,130]. This type of lesion is more common in patients suffering from acute MS or RRMS [131]. A large portion of these macrophages appears to exhibit a foamy appearance due to their myelin-loaded cytoplasm [132]. Furthermore, some of these active lesions exhibit myelinating activity, with the presence of macrophages containing myelin degradation in their cytoplasm [133].

As individuals living with MS progress through the disease, the frequency of lesions with mixed morphology of active and inactive characteristics increases. This heterogeneity, also known as slowly expanding lesions, has centers containing an inactive demyelinated site filled with myelin debris surrounded by activated macrophages and microglia around the rim and axonal injury observed in the active rim [134,135]. The slow expansion on these rims reflects ongoing demyelinating activity [67], whereas in patients with disease duration of more than 15 years or who have been diagnosed with progressive MS, inactive lesions are dominant [131]. In chronic lesions, demyelination, reactive gliosis, and the presence of dense fibrillary scar tissue can be observed between the transected axon space [108,136]. Furthermore, ongoing axonal damage can be identified within inactive lesions, highlighted by the disruption of the axonal transport [137]. Remyelination plaques are observed throughout all stages of MS, deemed shadow plaques, whereby axons are wrapped with new myelin sheaths. However, in chronic stages of MS, even though new myelin is present, these axons are still prone to damage due to variations of thicknesses between the inner and outer sheaths, specifically the thinner inner sheaths [138,139,140].

Interrogation of archived brain tissue derived from individuals who lived with SPMS has revealed the presence of B cell-like follicles in the meninges and the perivascular space, suggesting the possibility of trapped immune cells within the CNS after MS relapses in transition into progressive stages [141]. Compartmentalized inflammation in the progressive stage of MS is associated with cortical demyelination, slow expanding white matter lesions, diffuse injury in NAMW, and gray matter demyelination (for review, see [131,141,142,143,144]). These data may suggest that during progressive stages of MS, a requirement for future therapeutics is to reach inflammatory sites located behind the repaired BBB at an appropriate dose to resolve inflammation within the CNS.

## 3. Biomarkers of MS

Currently, there are limited biomarkers available to predict MS progression. One of the most highly researched areas of MS diagnostics need to include the elucidation of novel biomarkers that can clearly identify MS progression early for clinical management and to inform therapeutic efficacy [145,146,147,148].

### 3.1. Current and Developing Molecular Biomarkers

#### 3.1.1. Oligoclonal Bands

In MS patients, oligoclonal bands (OCBs) are used as molecular biomarkers for diagnosis and prognosis (For review, see [149]). OCBs consist of immunoglobulin G (IgG) and immunoglobulin M (IgM) produced intrathecally by plasma cells in the CNS (for review, see [150]). Their identification in cerebrospinal fluid (CSF) samples via isoelectric-focusing techniques on an agarose gel, followed by immunoblotting, is accompanied by paired blood sera to demonstrate intrathecal antibody production from differentiated B lymphocytes [151,152]. Although CSF-specific OCBs can be detected in more than 95% of MS patients (for review, see [153]), the presence of OCBs in the CSF is often correlated with the conversion from a clinically isolated syndrome (CIS) to the onset of MS [154]. Furthermore, CSF OCBs are not unique to MS, and the diagnosis still requires exclusion of other infectious and autoimmune diseases such as polyneuritis and optic neuritis (ON) [155,156]. Thus, CSF is the best representation of intrathecal CNS integrity, but to obtain CSF, an invasive lumbar puncture needs to be carried out [157,158]. Hence, improved diagnostic biomarkers with clinical assaying as well as advanced specificity are urgently required.

#### 3.1.2. Neurofilament Light Chain

Neurofilaments are integral structural components of the neuronal cytoskeleton, composed of three subunits: heavy chain (NfH), medium chain (NfM), and light chain (NfL) (for review, see [159]). They are found in axons, especially in long-projection axons, and they contribute to the axonal volume and mechanical strength [160,161,162]. Both NfL and NfH levels were shown to be elevated in the CSF of RRMS and progressive patients [163,164,165]. However, the role of NfH as a biomarker for MS progression is more underrepresented in biomarkers studies than NfL, which has a higher sensitivity in distinguishing between control and MS patients than NfH [166,167]. Recently, investigators assessed the potential of neurofilament light chain (NfL) as a diagnostic and prognostic biomarker for MS. NfL is the most abundant subunit; it is released from injured axons and becomes soluble in the CSF and serum (for review, see [168]). Neurofilaments are also released into the circulation during normal aging (for review, see [169,170]). With the development of a highly sensitive immunoassay such as the Simoa^®^ assay, the detection of NfL at femtomolar levels in the serum of MS patients is made possible and may assist with the clinical interpretation of MS progression [171,172,173], negating the need to extract NfL from the CSF in these patients.

Despite controversial relevance in neurological progression, recent clinical studies reported higher serum NfL levels in MS patients than in healthy individuals in relation to disease progression, potentially highlighting the detection of NfL levels as an ancillary diagnostic tool for MRI-detected lesions [174,175]. The increase in serum NfL was also shown to be associated with the loss of brain and spinal cord volume [174]. Experimentally, NfL levels are a robust detection marker of ascending paralysis at 18 days post-EAE immunization, whereby the NfL level increased 100-fold higher than the baseline level of healthy mice [176]. Malmeström et al. observed the highest level of NfL during acute relapses in RRMS patients, but the level declined within three months [164]. Furthermore, Salzer et al. discovered that RRMS patients with CSF NfL levels higher than 386 ng/L were more likely to convert to the secondary progressive phase of the disease compared to patients with NfL levels lower than 60 ng/L [177]. However, the change in the level of NfL and its association with disease progression has been inconsistent (for review, see [178]). In some studies, the NfL level of progressive MS patients was higher compared to RRMS patients [165,175,179], whereas others reported that the NfL level of progressive MS patients was lower than RRMS [180,181]. Most of the studies found that there was no significant difference in NfL levels between RRMS and progressive MS groups [147,182,183]. Therefore, more studies are required to examine the change in NfL levels in terms of MS progression.

Further experimental evidence identifying the potential clinical activity for NfL levels during therapeutic intervention included DMT glatiramer acetate, which resulted in a decrease in serum NfL levels of 81% compared to untreated EAE-induced mice [176]. These results indicate that NfL levels may serve as a prognostic marker during therapeutic intervention. However, Aharoni et al. detected that EAE induced by myelin proteolipid protein (PLP) immunization compared to myelin oligodendrocyte glycoprotein (MOG)_35-55_ displayed lower NfL levels at the same time point of clinical progression [176], suggesting that different myelin peptides induce variable neurodegenerative change, potentially making it difficult to set NfL limits for different MS histopathologically characterized lesions.

NfL can be safely, easily, and objectively measured; it is sensitive to neuroaxonal injury; and the level changes depending on the severity of the disease, making it suitable as a biomarker for the neurodegenerative process [145,184]. However, a major clinical disadvantage of using serum NfL levels as a biomarker is that the levels do not distinguish MS from other neurodegenerative diseases such as amyotrophic lateral sclerosis (ALS) and Alzheimer’s disease, which also exhibit elevated NfL serum and CSF levels [185,186,187]. Moreover, increases in CSF NfL levels can also be present in healthy subjects with increasing age, as shown by several clinical studies [169,188]. Indeed, healthy participants enrolled in a study to assess NfL levels between the ages of 18 and 70 years old displayed increased levels at a rate of around 2.2% per year of age [175]. Therefore, a better standardization specific to people living with MS who demonstrate neuronal tissue loss (taking into consideration age, sex, and other inclusion criteria) requires reforming to validate serum NfL and CSF NfL over the course of disease progression. Controversially, these criteria also need to take into account that some studies observing NfL-level ranges during MS often overlap with the ranges of NfL concentration in the control groups assessed (for review, see [189]), suggesting that there may exist a limited window of predictive efficacy for this biomarker.

#### 3.1.3. Chitinase-3-like-1

Chitinase-3-like-1 (CHI3L1) protein has also emerged as a potential biomarker for MS, and its expression is detected in active demyelinating lesions [190,191]. It is expressed mainly by reactive astrocytes in the CNS, with its expression upregulated during inflammation with gliosis [148]. The induction of its expression has been suggested to result from hyper-activated macrophages and microglia that produce inflammatory mediators that heighten CHI3L1 expression by astrocytes and further stimulate TREM2 [192]. An in vitro study utilizing the administration of CHI3L1 to cortical neurons exhibited neuronal function impairment, thus suggesting its neurotoxic properties [193]. The exact role of CHI3L1 requires further elucidation, but it is undeniably expressed within inflammatory events throughout the CNS and thus could be considered a surrogate biomarker that is indicative of MS microglia activity upon neuroinflammation in MS.

Elevated levels of CHI3L1 in CSF have been observed in CIS, RRMS, and progressive patients [148,192,194]. Patients progressing from CIS to RRMS had an increase in levels of CHI3L1 present in the CSF [148,195]. In RRMS, the expression of CHI3L1 correlated with loss of brain volume and advancement of disease activity [196]. However, another study indicated that an elevated level of CHI3L1 is significantly related to reduced cervical spinal cord volume but not to brain volume [194]. There is still controversy surrounding whether CHI3L1 levels could be detected either in CSF or serum to best reflect the disease course. The levels of CHI3L1 in MS patients were usually increased in the CSF, but another study observed increased serum levels of CHI3L1 in progressive MS patients [197]. This discrepancy could be explained by the different assays used to measure these levels [148,197]. Surprisingly, in the study by Cantó et al. there was no significant difference between CHI3L1 plasma levels in the remission and relapsing phase [197]. Nonetheless, studies have observed a correlation between CHI3L1 levels and NfL levels in the CSF, which were stronger in RRMS patients compared to SPMS and primary progressive MS (PPMS), further contributing to the debate about whether CHI3L1 is a suitable biomarker for progressive MS [198]. This study by Gil-Perotin et al. indicated that increased values of CHI3L1 were more prominent in progressive MS patients compared to NfL, which was more prominent in relapsing MS patients [198].

However, utilizing CHI3L1 still requires an invasive lumbar puncture, limiting its clinical utility. Its expression, like with NfL, has also been detected in other diseases, such as neuromyelitis optica [199] and Alzheimer’s disease [200]; therefore, it would not be suitable as a standalone biomarker but rather one that can be used alongside MRI to assess microstructural changes and diagnose the ongoing neurodegeneration of MS in patients. Further study is required to assess the role of CHI3L1 in chronic stages of MS to support the current promising results indicating the potential of CHI3L1 as a biomarker for neuroinflammation in MS.

### 3.2. Imaging Diagnostics as Biomarkers

#### 3.2.1. Magnetic Resonance Imaging

MRI is considered the gold standard for imaging diagnostics biomarkers in MS and a staple in diagnosing and identifying MS progression. The most clinically advantageous diagnostic measure for using MRI is the visualization of brain atrophy and neural tissue loss, commonly utilized to assess progression in MS patients. However, brain atrophy is not just limited to lesioned areas but is also visible in regions of NAWM [201,202]. An effective measurement to assess brain atrophy is the ventricular CSF (vCSF), which reflects the change in the overall volume of the brain [203]. It provides robust data, as vCSF is more commonly utilized across different MRI protocols and thus could possibly be the universal indicator of brain pathology in the clinical assessment of patients [203]. Nonetheless, brain atrophy changes have been observed to have a strong correlation with increased disability through the assessment of the objective expanded disability status scale (EDSS). Eijlers et al. assessed the annual deep gray matter and cortical atrophy rates in individuals with MS and found a strong correlation between brain atrophy and cognitive decline [204]. However, there are some factors that need to be considered that could affect brain atrophy measurements within the MS patient. For instance, an increase in brain volume, leading to increased atrophy, also occurs as a result of edema during inflammation [205,206]. Consideration must also be made where brain atrophy can be affected by factors such as age [207,208].

Although advanced MRI techniques such as diffusion tensor imaging MRI (DTI-MRI) and magnetization transfer MRI diagnosis are increasingly being used in MS research, they are still rarely employed in clinical practice [149,209,210]. Fluid-attenuated inversion recovery (FLAIR), T2-weighted MRI, and post-gadolinium T1-weighted scanning are the current mainstream diagnostic tools for MS [211,212,213]. T2-hyperintense lesions in MS patients are normally found in the periventricular, juxtacortical, infratentorial, and spinal cord regions [214]. However, T2-hyperintense lesions lack specificity to demonstrate the severity of the pathologic processes occurring in the lesions, such as inflammation, de/remyelination, gliosis, or axonal damage (for review, see [215]). Furthermore, T2 lesions show a weak correlation with clinical status, as measured by the EDSS [216,217,218]. Nevertheless, T2-weighted MRI could be a useful tool to reflect drug efficacy in clinical trials [219,220,221]. For example, Radue et al. identified that treating patients with fingolimod significantly decreased their T2 lesion volume compared to baseline [219].

In the T1-weighted pulse sequence, lipid-predominant structures such as myelin are displayed as bright spots, whereas water-predominant structures such as cortex are shown as dark anatomical regions. Indeed, it was recently demonstrated that demyelination and axonal degeneration reduced lipid content and increased the water content, resulting in the formation of hypointense signal areas on the T1 images [222]. Gadolinium-enhancing lesions are often associated with T1 hypo-intensity or “black holes,” which may indicate a combination of edema and demyelination [223,224,225]. Within 6 to 12 months, acute T1-hypointense lesions associated with gadolinium enhancement either convert back to T1-isointense lesions or remain chronic black holes [226,227,228]. DMTs such as fingolimod and glatiramer acetate have been shown to slow down the rate of disease development and repress the conversion from acute lesions to chronic black holes [229,230]. Compared to the progressive form of MS, gadolinium enhancement is less frequently seen in RRMS, which may correlate with the changeable immunoregulation of the innate and adaptive immune responses during the disease course [231].

However, progression maybe best monitored by the accumulation of extracellular iron. Dal-Bianco et al. suggested that gadolinium enhancement may not be able to reflect the characteristics of lesions robustly during both acute and chronic periods of MS [130]. Instead, utilizing iron imaging with 7-T MRI could be a feasible approach that may provide a more accurate depiction of particular lesions. From observation, there was reduced iron visibility in areas surrounding NAWM compared to early active lesions, indicating the stage where gadolinium enhancement is visible. Subsequently, iron accumulation in the CNS led to the formation of nodular iron spheroids, finally developing into FLAIR-hyperintense iron rims of microglia and macrophages in smoldering lesions, where gadolinium enhancement is barely detectable [232,233]. Iron-rimmed lesions were deemed more destructive and indicative of neurodegenerative progression and disease progression, as evidenced by increasing T1 hypointensity compared to non-iron-rimmed lesions and histological quantification of axonal damage through axonal spheroids, microglia activation, and Wallerian degeneration on areas correlating to these FLAIR-hyperintense regions [130]. In this longitudinal study of 33 MS patients up to 7 years, RRMS patients had significantly higher iron-rimmed lesions (17.8%) than SPMS patients (7.2%) [130]. Interestingly, there are studies that showed contrasting results as the consequence of variable clinical characteristics of patients, potential therapeutics that the patient was on, and whether this was done in vivo or conducted on post-mortem samples [131,234]. However, utilizing iron imaging with 7-T MRI to observe disease progression still requires more validation, especially with a more consistent and frequent follow-up on patients.

#### 3.2.2. Diffusion Tensor Imaging

DTI-MRI is principally based on tracking the movement of water within the CNS tissue and providing connectomics and network analysis of neural fibers. The main DTI parameters are fractional anisotropy (FA), axial diffusivity (AD), radial diffusivity (RD), mean diffusivity (MD), and apparent diffusion coefficient (ADC) (for review, see [235]). In MS patients, usually the FA and AD are decreased, whereas the RD and MD are increased compared to healthy controls (for review, see [235]). This is related to water mobility, which is negatively correlated with compact tissue integrity, such as myelin (for review, see [235]). 

Upon injury, the longitudinal movement of water along the axons is disrupted, which is suggested to cause a decrease in AD [236]. Nishioka et al. discovered that AD was reduced in the optic nerves of EAE-induced mice 4–8 weeks after the induction of the disease, along with reduced FA and increased RD [237]. Interestingly, a study by Naismith et al. presented evidence of decreased AD in individuals with acute optic neuritis; however, after a one-year follow-up assessment, AD had increased, as retinal ganglion cells possibly demonstrated recovery or repair [238]. Nevertheless, Budde et al. indicated that AD is a more specific marker of axonal damage in the EAE-induced mouse spinal cord compared to FA metrics [239]. However, Andersen et al. also found that in the corpus callosum body of secondary progressive MS patients there was a reduction in FA and an increase in RD [240]. The discrepancy in the change of DTI parameters has been attributed to the variable array of tissue samples used. 

Thus, to identify whether there is a correlation between DTI-MRI metrics and another promising NfL biomarker, Saraste et al. indicated an increase in serum NfL level accompanied by a variation in DTI-MRI metrics, where a decrease in FA was observed along with an increase in RD in NAWM in MS patients [187]. DTI-MRI techniques can also describe 3D neural fiber tracts by tomography, which, along with quantitative analysis of these DTI measurements, greatly enriched the interpretation of microstructural white matter damage. Moreover, DTI-MRI in combination with serum NfL level analysis could be a valuable monitoring tool for assessing the degree of the developing neurodegenerative processes during MS.

#### 3.2.3. Proton Magnetic Resonance Spectroscopy

Proton magnetic resonance spectroscopy (^1^H-MRS) is often used to investigate neurological disease and has a similar data acquisition process to conventional MRI, with only a few additional steps incorporated during the pre-scan (for review, see [241]). These pre-steps include improving the homogeneity in the magnetic field or “shimming,” suppressing the water signal, and choosing the appropriate MRS parameters or techniques (for review, see [242]). The principle underlying ^1^H-MRS is the signal detected from hydrogen protons utilized to measure the concentration of the metabolites in the CNS tissues of interest (for review, see [241]). In MS, the three metabolites of interest are creatine (Crn), N-acetyl aspartate (NAA), and choline (Cho). Crn is the marker of energy metabolism in the brain, and because of its stability, other metabolites are often shown as a ratio relative to Crn [243,244]. However, altered Crn levels in MS lesions have been detected by ^1^H-MRS [245,246,247,248]. Consequently, expressing other metabolites as a ratio relative to Crn may introduce more variability into the obtained measurements if the level of Crn is not stable [249,250]. In the MR spectrum, the peak of NAA can be used to analyze axonal integrity, whereas the peak of choline can be used to assess the cell-membrane metabolism (for review, see [10,251]). A lower peak of NAA in white matter signifies axonal damage, whereas a higher Cho peak is interpreted as higher membrane turnover, suggesting the occurrence of gliosis, demyelination, and remyelination events [10,252]. 

A decrease in the ratio of NAA/Crn in a given region of the brain indicates disrupted integrity of the tissue and thus has been suggested as a potential marker for neuroaxonal injury [253,254]. Aboul-Enein et al. observed a significantly reduced NAA/Crn ratio through ^1^H-MRS in patients with SPMS compared to RRMS patients in NAWM [255]. However, interestingly, this study observed no significant changes in NAA levels of RRMS patients compared to healthy individuals. In contrast, other studies have observed a decrease in this same parameter in NAWM and throughout the whole brain [256,257]. In addition, increases in Cho/Crn ratios in lesions from patients with RRMS and SPMS have been observed [258,259,260]. However, some studies refute this, having observed a decrease in Cho/Crn ratio [261,262]. The conflicting studies indicate that the different stages of MS and different regions being measured may affect the level of metabolites detected using ^1^H-MRS. Furthermore, the detection of limited metabolites through MRS does not reflect the full heterogeneity of MS lesions and thus could be utilized in support of other imaging biomarkers [255]. Additionally, when performing MRS, tissue or adjacent tissue with high susceptibility differences may result in the appearance of artifacts due to an non-uniform field (for review, see [241]).

#### 3.2.4. Optical Coherence Tomography

The vulnerability of the retinal ganglion cells and axons in MS leads to impairment of vision as a common symptom exhibited in MS patients. The onset of these visual symptoms could manifest as a result of optic neuritis, with 20% of these patients going on to develop MS, or as a result of lesions in their visual pathway (for review, see [263]). Although the exact mechanisms that bring about these degenerative outcomes require further elucidation, it is evident that there are morphological changes in retinal ganglion cells from the visual pathway as MS progresses [264,265]. These neurons and thin axons can be visualized through optical coherence tomography (OCT) imaging, which utilizes infrared light to create cross-sections to form a 3D image of the retinal layers. It is widely used to quantify the peripapillary retinal nerve fiber layer (pRNFL), which comprises ganglion cell axons before they coalesce to form the optic nerve [266]. Another measurement utilized clinically is the ganglion cell–internal plexiform layer (GCIP), which is the combination of the ganglion cell layer and the inner plexiform layer but is difficult to distinguish via imaging [267]. The accelerated rate of GCIP thinning was evident in early MS patients with ON, making GCIP a suitable marker for prognosis [268].

As a part of the CNS, the retina has the unique anatomical characteristic of its axons commencing from the anterior retinal nerve fiber layer being unmyelinated, allowing visualization of axonal degeneration without factoring in the myelin membrane disruption [269,270]. The thinning of pRNFL and GCIP has been associated with neurodegeneration and a reduction in ganglion cells compared to a healthy person or during normal aging [107]. Studies have identified a decrease in pRNFL layer thickness in MS patients with or without ON compared to healthy individuals [271,272,273,274,275]. In the same individuals, GCIP thickness decreased twice as fast in the MS patients compared to the healthy controls [275]. These studies were limited largely to only include RRMS patients, with a limited number of studies looking at OCT as an imaging biomarker in progressive MS patients. Hence, further investigations in a randomized control design should be explored.

Although OCT in a non-invasive manner can measure and monitor the axonal injury, there are some considerations that need to be clarified. Firstly, these studies have been conducted on defined populations and cannot be extrapolated to other ethnicities due to their differences in pRNFL levels and levels of thinning [276]. This could result from the prevalence of MS in different ethnicities, but further research is required to understand how epidemiological factors affect their morphological differences. Another factor with OCT is the variability from different software algorithms, as there are different versions of OCT imaging. Older studies mainly used time-domain OCT, which assesses the thickness of RNFL, whereas current studies utilize the spectral domain, which offers higher resolution and allows for the use of automated software algorithms to separate the layers of the retina [277,278,279]. Recently, more studies have used OCT angiography, which assesses the density of the retinal microvasculature [280,281], reducing the variability generated by technical errors. This suggests that the clinical use of OCT could be paired with other measures such as MRI assessments of brain atrophy and volume or visual acuity tests, or in correlation with the EDSS score for disease progression. However, the association of pRNFL with these factors requires further extrapolation, as studies show no correlation between these factors [282]. pRNFLs decreasing in thickness is also observed in other neurodegenerative diseases such as Alzheimer’s and Parkinson’s disease [283,284,285]; therefore, it may be insufficient as a specific diagnostic biomarker for MS but could be used as one to track neurodegeneration and monitor the disease progression with treatment.

#### 3.2.5. Positron Emission Tomography Scan

Positron emission tomography (PET) imaging allows for non-invasive assessment of MS pathological activities using radioligands that target the tissue or molecules driving lesional expansion. PET may be combined with tomography imaging in the presence of the 18 kDa translocator protein (TSPO) biomarker to detect microglial activation in MS, which provides an avenue to explore the pathophysiological outcomes of slow-burning lesions [286,287]. TSPO is an outer mitochondrial membrane protein, and high expression of TSPO is correlated with neuroinflammation and density of activated microglia [231,288,289]. The radioligands that have been used to target TSPO are [^11^C]PK11195 [290], [^11^C]PBR28 [291], [^18^F]FEPPA [292], and [^11^C]ER176 [293]. 

In a study conducted by Rissanen et al., an increase in the binding of [^11^C]PK11195 in lesions and NAWM in SPMS patients was reported, which is associated with ongoing microglial activity [294]. However, [^11^C]PK11195 has poor BBB permeability and high non-specific binding, whereas other tracers displayed heterogeneous binding to TSPO (for review, see [295]). Consequently, specific targets that can bind to the radiotracers need to be investigated. Examples of other targets currently under investigational trials include cannabinoid receptor 2, purinergic P2X7 receptor, and MERTK (for review, see [296]). Another limitation is that TSPO can also be expressed in activated astrocytes and endothelial cells [289], which decreases the specificity of evolving microglial activity. Furthermore, Nutma et al. discovered that in brain tissue from patients with MS, the increase in TSPO signal is associated with a higher density of microglia and astrocytes [286] but not the activation state of these cells, as seen in rodent studies [231,297]. This means that the interpretation of PET signals using TSPO depends on the design of the studies. 

Thioflavin T derivatives, such as Pittsburgh Compound-B (PiB), have also been used in PET imaging to detect amyloid pathology in Alzheimer’s patients due to its binding affinity for myelin, making such tracers potentially useful in assessing demyelination and even remyelination in MS [298,299]. Stankoff et al. presented evidence that the uptake of [^11^C]PiB was higher in NAWM compared to gray matter, and there was a lack of [^11^C]PiB uptake in the lesion sites of RRMS patients [300]. Similarly, Bodini et al. demonstrated dynamic changes of [^11^C]PiB binding in MS lesions over time using PET imaging, suggesting that it may demonstrate remyelination within MS lesions [299]. However, the tracers are derivatives modified from Aβ peptide tracers, where Aβ trafficking can influence the PET signals in white matter, thus influencing these measurements [301,302]. Pietroboni et al. suggested that these signals may not be derived from within white matter but rather from Aβ present in the CSF [302].

## 4. Potential MS Treatments

The current unmet medical need in treating progressive MS is limited due to CNS-specific pathological lesions separated in time and space with low efficacy of targeted agents to cross the BBB, especially in the secondary progressive phase, during which the BBB reseals. Once therapeutics can reach the CNS compartment with appropriate bioavailability, they must achieve efficacy by either limiting the pathogenetic mechanisms that propagate neurodegeneration or potentiate neurorepair. As such, phagocytosis of myelin debris that may contain inhibitory factors for remyelination could be key to improving neurorepair and limiting neurodegeneration during progressive MS. With the development of cellular therapies, the transplantable cells may remodel the inhibitory and inflammatory endogenous environment that ensues during MS. These novel therapeutic strategies may be designed to target ECM deposited during the evolution of pathogenic lesions or alternatively to cellular-based therapies targeting immune cells, glial cells, and neuronal responses that may initiate neurorepair.

### 4.1. Bruton’s Tyrosine Kinase Inhibitors: An Exciting New Development in Clinical Trials for People Living with Progressive MS

Recent developments in neurotherapeutic outcomes during secondary and primary progressive MS have emanated out of efficacy studies using Bruton’s tyrosine kinase (BTK) inhibitors to limit B-cell development and signaling [303,304]. The clinical rationale for utilizing selective BTK inhibitors during MS, which now include the CNS penetrant evobrutinib and tolebrutinib, relates to the changing conceptual focus on MS pathogenesis over the last decade, whereby B cells associated with intrathecal follicle-like structures may govern the brain atrophy associated with progressive MS [305]. Moreover, the efficacious clinical use of CD20 monoclonal antibody biologicals such as ocrelizumab and rituximab demonstrating B cell reduction may limit MS progression [306,307], suggesting that B cells may propagate neurodegenerative changes and are central to MS pathogenesis.

The complex interplay between the innate and adaptive immune mechanisms that govern chronic neuroinflammation (for review, see [308]) can involve intrathecally localized B cells perpetuating a cycle of inflammatory mediators. These mediators may include the B lymphocyte-associated factor (BAFF) and a proliferation-inducing ligand (APRIL) that may regulate the endogenous astrocytic responses commonly associated with progressive MS [309]. Alternatively, or simultaneously, the activation of the (constitutively or induced) BTK enzyme may regulate microglial cell-specific Fc receptor activation, which is critical for innate immune system demyelinating outcomes [310]. The new generation of selective BTK inhibitors such as tolebrutinib have been identified in phase 2b safety and efficacy studies to regulate the activity of gadolinium-enhancing lesions over a 12-week treatment period in a crossover trial design [304], clearly demonstrating reduced acute inflammation during MS. However, there has been some development with the possible clinical management of progressive MS whereby tolebrutinib is currently being assessed over a 12-week period for PPMS compared with ocrelizumab treatment (Clinical Trial Identifier NCT04544449) or a more protracted 6-month period in both primary and secondary progressive MS compared to a placebo (NCT04458051 and NCT04411641, respectively). Primary and secondary outcome measures do include brain volume imaging and neurodegenerative molecular biomarkers from baseline right up to study closure. These data may provide insights into pathogenic mechanisms that drive the chronic expanding lesion with interactions of the active intrathecal B-cell-like follicles with active microglia, or alternatively may suggest a totally lymphocyte-independent mechanism driving autoimmune demyelination [311].

### 4.2. Hyaluronan and Chondroitin Sulfate Proteoglycans

The ECM has been shown to play an important role in leukocyte activation and infiltration into the CNS. The increase in the production of hyaluronan in the ECM has been observed in both acute and chronic inflammatory sites, especially in the demyelinated lesions of MS and EAE [312]. Winkler et al. identified that hyaluronan binds to its receptor, CD44, on CNS endothelial cells to facilitate the migration of T lymphocytes into the CNS during EAE pathogenesis [313]. In support of this evidence, Kuipers et al. found that treatment of EAE-induced mice with 4-methylumbelliferone (4-MU), an inhibitor of hyaluronan production, limited T lymphocyte migration into the CNS parenchyma and reduced astrogliosis associated with disease progression [314]. They also demonstrated that 4-MU administration decreased the severity of EAE and promoted the differentiation of T lymphocytes to a Th2 phenotype and forkhead box P3 (FOXP^3^)+ regulatory T lymphocytes (Tregs) [314]. In addition to facilitating leukocyte migration, a high-molecular-weight (HMW) form of hyaluronan was produced by astrocytes in chronic demyelinated lesions of EAE-induced mice, and both HWM hyaluronan and hyaluronan fragments inhibited the maturation of oligodendrocyte progenitor cells (OPCs) into myelinated oligodendrocytes in demyelinated sites induced by lysolecithin (a neurotoxicant) [312,315]. Therefore, inhibiting the synthesis of hyaluronan and bioactive hyaluronan fragments may limit lymphocyte-mediated inflammation and promote remyelination in MS.

Another component of the ECM includes chondroitin sulfate proteoglycans (CSPGs), which play a central role in the formation of glial scars, can regulate lymphocyte migratory responses and remyelination potential with profound effects on axonal regeneration [316,317,318]. CSPG has been observed to be expressed in brain sections of MS patients and the EAE-induced animal model [318]. In a study conducted by Stephenson et al., CSPG expression was found to be elevated in the perivascular spaces of spinal cords during EAE, whereas decreased production of CSPG with fluorosamine treatment in this animal model led to inhibited leukocyte infiltration [318]. One of the degradation products of CSPG cleaved through the activity of chondroitinase ABC (chABC) is chondroitin sulfate proteoglycans-disaccharide (CSPG-DS). This cleaved product has also been shown by [319] to limit T-cell migration and cytokine production during the course of EAE. Additionally, Zhou et al. also found that CSPG-DS prevented EAE onset and limited the disease progression [320]. These studies showed that reducing the production of CSPG using fluorosamine or degrading CSPG using chABC prevented T-lymphocyte infiltration into the brain parenchyma, which reduced the inflammation in the CNS [320,321]. Furthermore, treatment with chABC reduced the misdirected growth of axons, allowing for regeneration in rats following nerve transection [322]. In this rat model of spinal cord injury (SCI), degrading chondroitin sulfate glycosaminoglycan chain with chABC facilitated the regeneration of the descending corticospinal tract axons, with functional recovery observed [323]. This functional recovery was also witnessed in a primate model of SCI treated with chondroitinase, where Rosenzweig et al. noted induced corticospinal axon growth in the gray matter [324].

Additionally, CPSGs have been shown to negatively influence the differentiation of OPCs in pathological conditions and limit remyelination during inflammatory demyelination. Siebert and Osterhout discovered that CSPG inhibited neurite outgrowth along with the differentiation of OPCs in vitro, but treatment with chABC reversed the inhibition [325]. In support of these findings, Lau et al. showed that decreasing the CSPG production with xyloside promoted remyelination in lysolecithin-treated mice [317]. Therefore, the inhibition of CSPG could attenuate the inflammatory environment to promote remyelination in an animal model of MS. However, clinically translating the use of chABC may present some challenges, such as its thermal sensitivity for its enzymatic activity, which has been shown to quickly decrease within three to five days at 37 °C [326]. Consequently, this has led to repeated delivery of chABC in animal models of SCI to the lesion sites to reach the deep regions of the spinal cord to observe therapeutic effects [327]. To try to overcome the thermal sensitivity, Lee et al. employed another method through thermostabilized chABC and a hydrogel-microtube scaffold-delivery system to provide sustained release of chABC locally in vivo in a rat model of SCI [326]. More studies need to be performed to investigate effective methods of blocking CSPG production, as it can provide future therapeutic benefits for MS patients.

In progressive MS patients, the most common type of lesion is the slowly expanding lesion with a hypocellular demyelinated center [140]. The glial scar and the loss of brain volume need to be replaced with cells, which can be transplanted or sourced from endogenous cells. However, aging affects the proliferation and differentiation capability of stem cells [328,329]. Since the majority of individuals living with progressive MS are usually of old age and only a few cells can be derived from endogenous stem cells, transplantation of inducible pluripotent stem cells (iPSCs) would be preferred. However, the glial scar that made up the lesion needs to be broken down first. Hyaluronan and CSPG are present in the glial scar as inhibitory factors. Thus, removal of CPSG and inhibition of hyaluronan, especially the high-molecular-weight form, can ameliorate scar formation by rendering the ECM more responsive to neurorepair and remyelination.

### 4.3. Cellular-Based Therapeutic Strategies

Cell-based therapies may be a promising approach to support the ongoing efforts of overcoming the lack of effectiveness of DMTs in patients with progressive MS. The concept of replacement of autoreactive and pathogenic cells and related output on signaling pathway regulation are of increasing interest in the research on progressive MS to eliminate autoreactive immune responses and enhance neurological recovery. Neural stem cells (NSCs) and autologous HSC-based transplantation has recently reached the clinical trial phase, opening the possibility of using cellular replacement as a regimen for MS patients.

#### 4.3.1. Neural Stem Cells

NSCs distribute in neurogenic stem cell niches, specifically enriched in the subventricular zone (SVZ) adjacent to the lateral ventricle in the forebrain and the subgranular zone (SGZ) of the dentate gyrus (DG) interface in the hippocampus (for review, see [330]). As mentioned earlier, remyelination in anatomical regions of the brain undergoing neurodegeneration is central in the treatment of MS, and this process requires the participation of endogenous NSCs, which can migrate to specific regions and differentiate into mature oligodendrocytes, restoring impaired neural function [331,332]. NSCs may treat MS through the modification of the endogenous microenvironment, converting from a neurotoxic to neurotrophic phenotype [333,334]. Moreover, transplanted NSCs may limit further damage to the site of injury by promoting cell replacement, immune regulation, nutritional support, and stimulating the differentiation of neural progenitor cells, alongside maintaining CNS homeostasis and optimizing the nervous system [335,336].

A major problem in achieving neurorepair through NSC transplantation during MS is to resolve the pro-inflammatory microenvironment and repopulate the parenchymal functional cells at the lesion site. However, excessive and disorganized ECM, apoptotic cell debris, and pro-inflammatory cytokines can affect the viability, differentiation, and migration ability of NSCs to the lesion site [337,338,339]. At present, several studies have confirmed that different in vitro induction methods can enhance the effect of NSC transplantation in MS. Moore et al. identified that estrogen receptor beta (ERβ) agonists can be used to enhance oligodendrocyte differentiation in EAE and to improve CNS myelin repair in mice along with improved clinical outcomes [340]. However, Imamura et al. showed in recent studies that donepezil, an approved treatment for AD, induced the expression of myelin-related genes and further stimulated the differentiation of NSC derived from iPSCs into oligodendrocytes through the ER signaling pathway, thus further improving myelin regeneration in the CNS [341]. In addition, some metabolites found in the human body regulated the effect of NSC transplantation. In vivo experiments also demonstrated that transplanted NSCs exerted anti-inflammatory action by scavenging succinate through the SUCNR1 signaling pathway, leading to the production of prostaglandin E2 [342]. Moreover, succinate has been shown to activate the SUCNR1/GPR91 signaling pathway in macrophages, promoting the polarization of macrophages to a pro-inflammatory phenotype and production of pro-inflammatory IL-1β [343]. Consequently, the uptake of succinate by NSCs resulted in the induction of prostaglandin E2-dependent anti-inflammatory effects and less succinate available to activate SUCNR1/GPR91 signaling pathway in macrophages [342]. When EAE-induced mice were transplanted with Sucnr1^−/−^ NSCs, microglia failed to convert to an anti-inflammatory phenotype, and only a slight recovery of behavioral outcomes was observed [342].

Bone marrow-derived NSCs and those derived from the SVZ have been shown to exert almost the same therapeutic effects in the EAE preclinical models, and interestingly, the BM-derived NSCs have similar morphological characteristics and similar capability of differentiating into neurons and glial cells to SVZ-derived NSCs [344]. Xie et al. injected bone marrow-derived NSC transfected with TGF-β1 into mice with EAE via the tail vein and transduced bone marrow-derived NSC inhibited Th1 and Th17 populations, promoting the production of immunosuppression through Treg cells and cytokine IL-10 from the periphery, thereby transforming microglia from a classical to an alternative pathway [345].

These results indicate that genetically modifying NSCs can be more effective at inhibiting clinical severity, inflammation, and demyelination in the CNS of mice. In addition, these investigations reported that the total number of neurons and oligodendrocytes in the CNS was significantly increased in the TGF-β1-transfected NSC transplantation group compared with the control group injected with normal saline [345]. However, there was no significant difference between the two groups treated with non-transfected NSCs when compared with the transfected NSC transplantation group. This suggests that TGF-β1 does not alter the proliferation and differentiation of NSCs, and thus, the more indicative markers for genetic modification of stem cells should be further identified.

A demonstration of remyelination of the CNS in EAE-induced male mice and an increased number of CD4^+^CD25^+^FoxP3^+^ Tregs were observed following transplantation of human NSCs into mice at the chronic stage of the disease [346]. Compared with the control group of mice induced with EAE, the neuroinflammatory response of mice treated with human NSC transplantation was significantly reduced [346]. Harris et al. evaluated the safety and tolerability of autologous bone marrow-derived NSCs for the treatment of 20 patients with MS in a phase I clinical trial [347]. In the follow-up 24 months post-transplantation of 20 participants, no new lesions were found on the T2 MRI and no severe adverse events were reported, serving as the evidence of the safety and tolerability of transplanted autologous bone marrow-derived NSC transplantation for MS in the short term [347].

#### 4.3.2. Autologous Hematopoietic Stem Cell Therapy (AHSCT)

HSC transplantation has been shown to be a potential treatment for ameliorating neurological deficits by promoting neural regeneration and functional recovery [348,349]. The current methodologies for transplantation include AHSC populations extracted from patients and reconstituted outside of patients’ bodies, characterized by high mobilized capability and low autoreactivity. Recently, a randomized controlled study (NCT00273364) reported the significant effect of nonmyeloablative HSCT on extending the time to disease progression in patients with RRMS compared with conventional DMTs [350]. More importantly, no death occurred and no adverse events over non-hematopoietic toxicity grade 4 were observed, confirming the safety, feasibility, and durability of AHSCT on MS patients [350]. The low incidence of transplant-related adverse events may be contributed to by the more specific candidate inclusion criteria. In the aging population, due to the concern of lower effectiveness of HSCs with functional decline (decreased capability of self-renewal and pluripotency) and the high risk of mortality or complications resulting from AHSCT, MS participants under 60 years old and with a relapsing–remitting stage duration of less than 5 years are recruited in most ongoing AHSCT clinical trials.

Gene modification techniques applied to cells (especially stem cells) before cellular-based transplantation is an ideal strategy that enhances functional cell survival and promotes CNS regeneration in the model of neurodegenerative diseases [351,352,353]. Currently, lentivirus (LV)-modified HSCT has been examined in several clinical trials majorly targeting immune, neurological, and genetic diseases such as human immunodeficiency virus (HIV) infection (NCT00002221), cerebral adrenoleukodystrophy (ALD) (NCT01896102), Krabbe Disease (NCT04693598), adenosine deaminase (ADA) deficiency (NCT01380990), X-linked severe combined immunodeficiency (SCID) (NCT03601286), and sickle cell disease (NCT02140554). Some debates continue about mutagenesis resulting from virus-carried genes integrated into HSCs. However, in the recent clinical trial that utilized gene therapy with HSCT for ALD, 88% of patients receiving genetically modified HSCT demonstrated stable long-term hematopoietic reconstitution with modified HSCT maintained in proportion to the patient’s original immune lineage [354]. Additionally, X-linked SCID patients transplanted with autologous LV-transduced HSCs showed immune reconstitution, and the treatment appeared to be safe [355]. Besides using viral vectors to genetically modify the HSCs, CRISPR-Cas9 is another gene-editing technology that was reported to be a safer alternative [356].

A signaling cascade that may govern neurodegenerative change within the CNS has been well established to involve the inhibitory-signaling cascade induced by myelin-associated inhibitory factors (MAIFs). These MAIFs include Nogo-A, which has the strongest inhibiting effect on CNS neuronal and myelinated oligodendroglial lineage cells through binding to Nogo receptor 1 (NgR1) (for review, see [357]). NgR1 and its coreceptors transmit downstream signaling that involves RhoA/Rho-associated coiled coil-forming kinases (ROCK) and CRMP2, which elicit profound inhibitory neurite outgrowth and axonal growth and can thereby limit synaptic plasticity [358,359]. Accordingly, the NgR ectodomain can be used as a decoy to recognize and bind to myelin debris that harbors the target antigen such as Nogo-A on their surfaces and further reduce the activation of the inhibitory downstream pathways related to axonal degeneration. NgR(310)ecto-Fc, the soluble decoy fusion protein, is capable of binding to MAIFs and driving the therapeutic effect on animal models of glaucoma and SCI [360,361]. Therefore, the NgR(310)ecto-Fc fusion decoy protein may be a plausible self-activating vector transduced in HSCs to re-establish the immune system and could be expected to become a novel cell-based therapeutic for patients with immune-related neurodegenerative diseases.

HSCs can differentiate into various immune cells such as T cells, B cells, and macrophage (for review, see [362]) and HSC-derived immune cells are capable of crossing into the CNS [363]. Using gene-editing technology, HSCs can then be utilized as vehicles to deliver the NgR(310)ecto-Fc decoy protein from the periphery to the target lesions in the CNS. As mentioned previously, macrophages and microglia are capable of phagocytosing myelin debris, but there seems to be insufficient clearance of this debris in patients with MS [364]. This myelin debris is well understood to be inhibitory to axonal repair and the differentiation of OPCs to mature oligodendrocytes, which remyelinate the axons [365,366]. NgR(310)ecto-Fc may bind to Nogo-A concentrated in myelin debris, and the Fc portion of the fusion protein can bind to the Fc receptor localized on the cell membranes of macrophages/microglia, facilitating the clearance of inhibitory myelin debris. Furthermore, NgR(310)ecto-Fc can also inhibit myelin-mediated axonal degeneration by blocking Nogo-A/NgR signaling [365]. Our recent data demonstrated significant remyelination and axonal repair during EAE following transplantation with NgR(310)ecto-Fc-transduced HSCs [367].

## 5. Conclusions

The major cellular players in MS consist of T cells, B cells, macrophages, astrocytes, microglia, and oligodendrocytes that undergo reactive changes when exposed to pathological conditions in the CNS. Although diverse immunomodulatory DMTs are being investigated, they mainly target the T and B cells and aim to resolve inflammation during the acute stage of RRMS. Moreover, there are only a few approved DMTs for patients with progressive MS who have suffered from permanent neuronal and behavioral dysfunction due to extensive demyelination and neurodegeneration. Many studies focus on the clearance of myelin debris and the removal of inhibitory substrates in the glial scar to promote neuroregeneration, prompting the adaptive immunity- and glia-mediated response to facilitate the neural repair process. 

Histopathologically, acute demyelinating inflammatory lesions substantially develop into chronic active and chronic inactive plaques, and these chronic lesions are typically observed in the post-mortem tissue sections of SPMS patients. The distribution of immune and glial cell populations within chronic plaques in progressives MS is vastly different from acute lesions in RRMS. The presence of demyelinated axons and reactive gliosis from activated astrocytic and microglial interactions contributes to the formation of glial scars, which are observed in chronic MS plaques. Classical MS-like disease models, including EAE and cuprizone toxin models, have similar immunopathology and glial pathology with MS, and these models are being used to investigate the complexity of cellular responses during neurodegeneration. Nonetheless, several limitations, such as the insufficient manifestation of MS progression and pathophysiological hallmarks in EAE and cuprizone toxin models, still exist. Hence, a newly developed animal model needs to be investigated, particularly linking inside-out and outside-in mechanisms to outline a more comprehensive profile of MS cellular pathology and pathogenesis. A recent study using a newly developed MS-like model, defined as cuprizone autoimmune encephalitis (CAE), reported that both endogenous and peripheral immune responses can be activated during lesion formation and demyelination [368,369]. This finding provides evidence for the inside-out mechanism in progressive MS that subclinical demyelination triggers the activation of endogenous immune cells in the CNS and adaptive immune response, thus contributing to further axonal degeneration. Moreover, this animal model may provide an explanation regarding why DMTs are ineffective against the progressive stage of MS, as DMTs focus less on repairing the damage elicited by peripheral inflammatory cells and endogenous glial cells in the CNS.

One issue in translational research of MS is that there are limited molecular and cellular biomarkers available for monitoring the disease severity and examining the effectiveness of the therapies over the course of MS. As a consequence of inconsistent and unpredictable disease progression in MS patients, it is difficult to find a biomarker with well-evaluated validity and clinical relevance. From the visualization perspective, advanced MRI techniques provide several imaging biomarkers through the assessment of brain volume and the thickness of the cortical cortex, which have helped to establish the microstructural changes in the CNS for MS patients. However, fluctuations in parameters obtained from MRI techniques in different MS patients highlight the potential risk of results that are rarely reproducible. The potential strategy of combining molecular biomarkers with imaging techniques to investigate either new therapeutics or disease progression monitoring provides a feasible way to obtain a more accurate analysis.

Another important consideration to improve the clinical outcomes of MS patients is to investigate potential therapeutics. Stem cell transplantation combined with genetic modification techniques has become a promising option for treating autoimmune diseases and immunodeficiency disorders. This therapeutic strategy not only replaces the pathogenic immune cell populations with healthy cells but also produces therapeutic proteins by genetic-editing techniques on transplanted cells. The delivery of decoy fusion protein NgR(310)ecto-Fc via genetically modified HSCs directly into the lesions can inhibit neurodegeneration and improve neurorepair, potentially offering long-term neuroprotection after the progressive stage of the disease. The repair can also be assessed using existing molecular and imaging biomarkers. In the future, genetically modifying HSCs with CRISPR-Cas9 to deliver NgR(310)ecto-Fc may improve the safety of the treatment and possibly offer a new therapeutic option for progressive MS patients.

## Figures and Tables

**Figure 1 ijms-24-11112-f001:**
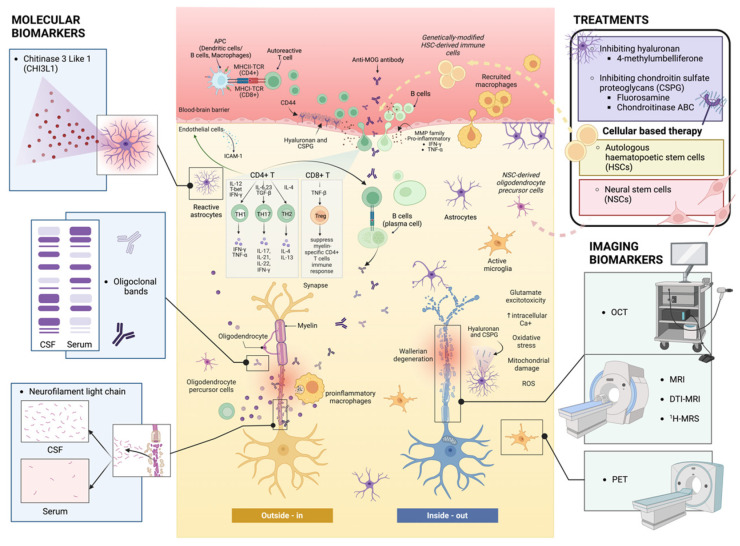
Outside-in and inside-out mechanisms during the pathogenesis of MS. The outside-in mechanism illustrates the infiltration of reactive CD4+ and CD8+ T cells, along with B cells, that travel from the periphery, across the BBB, and into the CNS. These cell types target and attack myelin-associated epitopes on myelinated axons, secreting a myriad of pro-inflammatory cytokines, which leads to neural cell dystrophy and glial reactive changes to promulgate neurodegeneration. Conversely, the inside-out mechanism employs the concept of Wallerian degeneration and axonal and oligodendrocyte dystrophy, whereby remyelination failure in MS is mediated by astrocytes and peripheral/resident immune cell responses through slowly progressive lesions. CSF: cerebrospinal fluid; APC: antigen-presenting cell; TCR: T-cell receptor; MOG: myelin oligodendrocyte glycoprotein; MMP: matrix metalloproteinase; ICAM: intercellular adhesion molecule; ROS: reactive oxygen species; OCT: optical coherence tomography; MRI: magnetic resonance imaging; DTI: diffusion tensor imaging; ^1^H-MRS: proton magnetic resonance spectroscopy; PET: positron emission tomography (figure created with BioRender).

## Data Availability

Not applicable.

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
