# Peer review of "The Heterogeneous Multiple Sclerosis Lesion: How Can We Assess and Modify a Degenerating Lesion?"

_ijms, 2023, doi:10.3390/ijms241311112_

Round 1
Reviewer 1 Report
I think a graphical abstract would be very useful to be included, as there are a lot of notions included in a narrative manner.
Author Response
We thank the Reviewer for their recommendation of our manuscript and have incorporated a graphical abstract in the revised verion.
Reviewer 2 Report
I thank the authors for the interesting review of demyelinating lesions in progressive MS, as well as molecular and imaging biomarkers of this pathology. I commend the authors for describing this critical and timely issue. The paper is interesting and well-written; however, I would like to highlight some issues that merit revision.
Many sentences are overly complex, making them difficult to understand. Sometimes there are too many unnecessary details.
In the section "Pathogenesis" in several places the influence of the drug "cuprizone" is mentioned, it is necessary to report about the drug itself in more detail.
Lines 77-78. The sentence has no context.
Paragraph 113-131. is not related to the content of the previous text
Line 134. please give a more specific title for this item, e.g. “hypothetical causes of MS”
Line 147. It is not clear what the whole genome analysis showed. Reword the sentence
Lines 213-225. Extra details are given, such as information on the classification of monocytes. The phrase “the green fluorescent protein (GFP) level did not accurately reflect the actual CX3CR1”, with a high probability, was copied from the article [59], however, without context, it does not carry an informational load. The main theses from the cited literature should be briefly described.
Lines 218-222 Should be decoded: CX3CR1, CX3CL1, TREM2
Lines 229-236. Don’t clear. What cells express the markers CD40, CD206, CD163. And what follows from this conclusion?
Lines 252. Please tell more about antibodies that recognize myelin sheath proteins. Mention should be made of catalytically active antibodies in MS.
Lines 296-303. The meaning of these proposals is not clear. Please formulate your idea more clearly. Information about gene expression should be moved to the paragraph on transcriptomic analysis (318-333)
Lines 304-306. Please reformulate the sentence.
Lines 309-311. Please list the phenotype of microglia in white and gray matter in healthy donors for comparison.
Line 347. Please describe "reactive astrocytes" in more detail.
Line 458. Which assay visualizes oligoclonal bands in cerebrospinal fluid?
Lines 491-491. The proposal is similar to what was written above.
Lines 736-743. The results of several studies can be combined into one specific conclusion.
Lines 823-831. Please provide examples of Bruton's tyrosine kinase (BTK) inhibitors
Lines 833-853. The paragraph is very hard to read. Please formulate the essence of the question.
Line 953. Please use italics for “in vitro”.
Author Response
We thank the reviewer for their insightful suggestions to improve the readability of the manuscript. We have incorporated their suggestions into the manuscript below:
"I thank the authors for the interesting review of demyelinating lesions in progressive MS, as well as molecular and imaging biomarkers of this pathology. I commend the authors for describing this critical and timely issue. The paper is interesting and well-written; however, I would like to highlight some issues that merit revision.
Many sentences are overly complex, making them difficult to understand. Sometimes there are too many unnecessary details.
In the section "Pathogenesis" in several places the influence of the drug "cuprizone" is mentioned, it is necessary to report about the drug itself in more detail." Answer (A): We have now added lines 195-200.
Lines 77-78. The sentence has no context. (A): We have now modified this sentence that is now lines 77-80.
Paragraph 113-131. is not related to the content of the previous text. (A): We have now added a subsection title for this paragraph on line 115.
Line 134. please give a more specific title for this item, e.g. “hypothetical causes of MS”. (A): We have now modified the subsection title for this paragraph on line 138.
Line 147. It is not clear what the whole genome analysis showed. Reword the sentence. (A): We have now amended the sentences on lines 151-155.
Lines 213-225. Extra details are given, such as information on the classification of monocytes. The phrase “the green fluorescent protein (GFP) level did not accurately reflect the actual CX3CR1”, with a high probability, was copied from the article [59], however, without context, it does not carry an informational load. The main theses from the cited literature should be briefly described. (A): We have now incorporated the sentence on lines 231-233 to provide context to the experimental evidence discussed.
Lines 218-222 Should be decoded: CX3CR1, CX3CL1, TREM2. (A): We have now added the definition of the forementioned abbreviations on lines 226, 237 and 240.
Lines 229-236. Don’t clear. What cells express the markers CD40, CD206, CD163. And what follows from this conclusion? (A): We have now included on lines 248, 249 and 253-260 respectively.
Lines 252. Please tell more about antibodies that recognize myelin sheath proteins. Mention should be made of catalytically active antibodies in MS. (A): We have now added a description of Abzymes on lines 276-287.
Lines 296-303. The meaning of these proposals is not clear. Please formulate your idea more clearly. Information about gene expression should be moved to the paragraph on transcriptomic analysis (318-333). (A): We have now restructured the entire paragraph from lines 328-352.
Lines 304-306. Please reformulate the sentence. (A): We have now re-written the sentences from lines 376-384.
Lines 309-311. Please list the phenotype of microglia in white and gray matter in healthy donors for comparison. (A): We have now added this description on lines 343-344.
Line 347. Please describe "reactive astrocytes" in more detail. (A): We have now added the description on lines 414-420.
Line 458. Which assay visualizes oligoclonal bands in cerebrospinal fluid? (A): We have now added the description on lines 533-534.
Lines 491-491. The proposal is similar to what was written above. (A): We have now modified the sentence on lines 558-559.
Lines 736-743. The results of several studies can be combined into one specific conclusion. (A): We have now modified the sentence on lines 810-816.
Lines 823-831. Please provide examples of Bruton's tyrosine kinase (BTK) inhibitors. A): We have now included examples of BTKIs on lines 899-900.
Lines 833-853. The paragraph is very hard to read. Please formulate the essence of the question. A): We have now modified the paragraph on lines 907-931.
Line 953. Please use italics for “in vitro”. A): We have now modified this on line 1031.
Reviewer 3 Report
Authors accomplished an extensive review about the more recent findings in the field of pathogenesis, diagnosis and candidate treatments of Progressive Multiple Sclerosis. The references are up to date and all the most important issues related to the topic are included.
I have no additional suggestions for the authors.
Author Response
We thank the reviewer for their recommendation of our manuscript
Reviewer 4 Report
In this review, Ellen et al. comprehensively review a number of topics related to multiple sclerosis. It is quite comprehensive, and detail oriented on a number of number of far reaching topics.
My major reviewing comment is in fact that this article is too comprehensive. It feels as though I am reading an amalgam of a number of papers. This is illustrated by the number of references (363!). I would highly recommend the authors focus the review on a single, more cohesive topic. Give the molecular focus of the journal, perhaps a review of the molecular biology of a multiple sclerosis plaque as well as revision to better reflect the focus of the manuscript.
As a multiple sclerosis physician scientist, I think the focus on a "progressive" lesion is somewhat misguided. I would claim that a lesion isn't necessarily "progressive," but a patient can be. Perhaps the term "smouldering" lesion would be more applicable.
There a handful of minor English revisions that I would be happy to address in follow up.
Author Response
We thank the reviewer for their suggestions to improve our manuscript. We address their comments below:
"My major reviewing comment is in fact that this article is too comprehensive. It feels as though I am reading an amalgam of a number of papers. This is illustrated by the number of references (363!). I would highly recommend the authors focus the review on a single, more cohesive topic. Give the molecular focus of the journal, perhaps a review of the molecular biology of a multiple sclerosis plaque as well as revision to better reflect the focus of the manuscript." Answer (A): Our review covers a description of the cellular and molecular mechanisms that govern the pathogenesis of MS lesions in time and space. The review provides the reader with clear context to the behaviour of expanding lesions, current methods of validating neural injury outside of the EDSS clinical measure, and how these developing biomarkers can be utilized during trials of novel therapeutics that can modulate the cellular and molecular reactions to injury covered throughout the review. If we were to condense the entire manuscript it would leave gaps for the reader that would have no specific context.
As a multiple sclerosis physician scientist, I think the focus on a "progressive" lesion is somewhat misguided. I would claim that a lesion isn't necessarily "progressive," but a patient can be. Perhaps the term "smouldering" lesion would be more applicable. (A): We thank the reviewer for their accurate description that the lesion should not be defined as progressive but only seen in individuals living with progressive MS. We have hence amended the title to depict this.
Round 2
Reviewer 2 Report
The new additions to the manuscript made a big difference. The quality of the paper had improved, and all my questions were addressed. No more comments.